# Durability of Immune Response to ChAdOx1-nCoV-19 Vaccine in Solid Cancer Patients Undergoing Anticancer Treatment

**DOI:** 10.3390/vaccines10101662

**Published:** 2022-10-05

**Authors:** Passakorn Wanchaijiraboon, Nattaya Teeyapun, Nussara Pakvisal, Panot Sainamthip, Thiti Susiriwatananont, Nicha Zungsontiporn, Nungruthai Suntronwong, Preeyaporn Vichaiwattana, Worata Klinsawat, Nasamon Wanlapakorn, Suebpong Tanasanvimon, Virote Sriuranpong, Yong Poovorawan, Sutima Luangdilok

**Affiliations:** 1Phrapokklao Cancer Center of Excellence, Phrapokklao Clinical Research Center, Phrapokklao Genomic Laboratories, Phrapokklao Hospital, Mueang District, Chanthaburi 22000, Thailand; 2The King Chulalongkorn Memorial Hospital, Bangkok 10330, Thailand; 3Department of Pharmacology, Faculty of Medicine, Chulalongkorn University, Bangkok 10330, Thailand; 4Center of Excellence in Clinical Virology, Department of Pediatrics, Faculty of Medicine, Chulalongkorn University and The King Chulalongkorn Memorial Hospital, Bangkok 10330, Thailand; 5Conservation Ecology Program, School of Bioresources and Technology, King Mongkut’s University of Technology, Bangkok 10150, Thailand; 6Division of Medical Oncology, Department of Medicine, Faculty of Medicine, Chulalongkorn University, Bangkok 10330, Thailand; 7Department of Biochemistry, Faculty of Medicine, Chulalongkorn University, Bangkok 10330, Thailand

**Keywords:** COVID-19 vaccine, Omicron, ChAdOx1, cancer, immunogenicity, SARS-CoV-2

## Abstract

There are limited data available about the durability of the immune response after administration of the widely used adenovirus-vectored ChAdOx1-nCoV-19 vaccine in cancer patients. This prospective longitudinal observational study analyzed follow-up data of immunogenic responses 12 weeks after the second dose of the ChAdOx1-nCoV-19 vaccine in 290 oncological patients compared to healthy controls. The study aimed to assess the persistence of the humoral immune response three months after the second dose, and omicron neutralization was also evaluated. Three months after completion of the second vaccine dose, the geometric mean titer of SARS-CoV-2 binding total Ig statistically decreased by 42% compared to those at 4 weeks, and was lower than that of the healthy control. Six percent of patients became seronegative for anti-RBD total Ig. Only 5% (2 of 40 samples) tested positive for surrogate neutralization against SAR-CoV-2 Omicron BA.2. Across different therapy types, a waning in immunogenicity was observed within three months after the second dose of the ChAdOx1 nCoV-19 vaccine, rendering it insufficient at that point to protect against the SAR-CoV-2 Omicron BA.2 variant.

## 1. Introduction

Oncologic patients undergoing anticancer treatment have a higher risk of developing severe symptoms and other adverse outcomes from COVID-19 than non-cancer populations [1,2,3,4]. A reduction in SARS-CoV-2 infection (5% vs. 0.4%) and mortality (0.7% vs. 0.08%) was demonstrated in cancer patients who received two doses of a COVID-19 vaccination (either BNT162b2, mRNA-1273 or ChAdOx1-nCoV-19) compared to those received one dose [5]. Several clinical factors, including male sex, older age, poor Eastern Cooperative Oncology Group (ECOG) performance status, chronic use of steroids, and active chemotherapy, were reported to be correlated with a reduction in IgG titers [6,7,8], since these immunocompromised patients were not well-represented during the early phase of clinical trials. Data on the durability of the humoral immune response in cancer patients after vaccination are limited. Previous studies reported reduced immunogenicity following two doses of BNT162b2 vaccine in solid cancer patients who were treated with chemotherapy; moreover, the seropositive proportion in the group of oncologic patients remained high after the second vaccination dose [9,10]. Little is known about the response of such patients to a viral vector-based vaccine, which is the primary vaccine available for cancer patients in Thailand. Our previous study reported reduced immunogenicity in solid cancer patients who received two doses of ChAdOx1-nCoV-19, particularly in those who received chemotherapy and immunotherapy [11,12]. A better understanding of the long-term immune response against SARS-CoV-2 will enable us to better predict when the next vaccine booster should potentially be required, and to assess vaccination efficacy in cancer patients and other priority groups.

This longitudinal study aimed to assess the persistence of the anti-SAR-CoV-2 spike protein receptor binding domain (anti-RBD total Ig), including its decay rate and surrogate neutralization (sVNT) against the Omicron BA.2 variant, following viral vector-based vaccines in patients undergoing varying types of cancer treatment. In order to guide priorities and provide a rationale for a booster dose vaccination in these vulnerable patients, quantitative post-vaccination immune response data are required to inform public health decisions.

## 2. Methods

### 2.1. Study Design and Participants

This study reported the follow-up data, namely the immunogenicity at 12 weeks post second dose of the ChAdOx1-nCoV-19 vaccine, in solid cancer patients who were previously immunized with a primary series of ChAdOx1-nCoV-19 vaccines at two cancer centers in Thailand: King Chulalongkorn Memorial Hospital (KCMH), and Phrapokklao Hospital (PPK). The initial immunogenicity results were previously reported [11,12]. The study was approved by the Institutional Review Board of the Faculty of Medicine, Chulalongkorn University (No. 486/64), and the Chanthaburi Research Ethics Committee/Region 6 (CTIREC) (No. 044/64).

The objective was to assess the durability of humoral immunity against SARS-CoV-2 at 12 weeks after the second vaccination, in comparison with healthy controls. Patients who acquired SARS-COV2 infections or were positive for anti-nucleocapsid antibodies were excluded from the durability analysis. The effects of different anticancer treatments on the immune response, as well as other clinical factors that were associated with reduced immunogenicity, were also explored.

### 2.2. Study Outcomes

The endpoint was the anti-RBD total Ig, 12 weeks after the second vaccination in oncologic patients who were previously administered with two doses of the ChAdOx1-nCoV-19 vaccine.

### 2.3. Healthy Individuals for Comparison

Data from 84 of 90 healthy individuals, who were previously recruited in a prospective cohort study by the Center of Excellence in Clinical Virology, Department of Pediatrics, Faculty of Medicine, Chulalongkorn University, and the King Chulalongkorn Memorial Hospital since March 2021, were compared to our oncologic patients [13]. Healthy individuals were administered two doses of the ChAdOx1-nCoV-19 vaccine, 10 weeks apart. Compared to the cancer cohort, we performed a serological assessment four weeks after the second dose, and twelve weeks after the second dose. Five out of ninety patients were excluded, because they failed to complete the follow-up schedule, and one patient showed a positive result for anti-nucleocapsid antibodies.

### 2.4. SARS-CoV-2 Serological Assessment

Total immunoglobulin specific to the receptor-binding domain (RBD) of the SARS-CoV-2 spike (S) protein (anti-RBD total Ig) was measured using the Elecsys^®^ (Roche Diagnostics, Basel, Switzerland) SARS-CoV-2 S assay. This assay uses a recombinant protein that represents the RBD of the S antigen in a double-antigen sandwich assay format, which favors the detection of antibodies against SARS-CoV-2 by Cobas e411 immunoassay analyzers. The serum samples were processed following the manufacturer’s protocol. A SARS-CoV-2 total antibodies concentration below 0.8 U/mL was considered to be seronegative, and a concentration titer above 210 U/mL was defined as being an adequate response for COVID-19 convalescent plasma, as defined by current FDA emergency use authorization guidelines [14]. The Elecsys^®^ Anti-SARS-CoV-2 S immunoassay reported the total antibody titer in U/mL units. In order to convert to the WHO international standard for anti-SARS-CoV-2 immunoglobulin, 1 U/mL was divided by 0.972, which is equivalent to 1 BAU/mL.

Anti-nucleocapsid (N) IgG was measured using enzyme-linked immunosorbent assays (Abbott Diagnostics, Abbott Park, IL, USA). The anti-N IgG was reported as the optical density (OD) unit of a sample per calibrator (S/C) ratio, with a cut-off value ≥ 1.4 scored as a positive result.

### 2.5. Surrogate Neutralization for Omicron

In order to evaluate the neutralizing activity against the Omicron BA.2 variant in the cancer cohort 3–4 months after the second dose, an ELISA-based surrogate virus neutralization test (sVNT) was performed using the cPass SARS-CoV-2 neutralizing antibody detection kit (GenScript, Piscataway, NJ, USA), and Omicron RBDHRP (Z03730) with the sVNT kit, following the manufacturer’s protocol. Forty serum samples in high-ranking antibody titers were chosen for an initial evaluation. A positive result was defined to be a percentage of inhibition that was more than 30%.

### 2.6. Statistical Analysis

Descriptive statistics were used to describe the demographic data. Paired and unpaired *t*-tests were applied to compare the log10 of anti-SARS-CoV-2 total antibodies in dependent and non-dependent variables, respectively. The Kruskal–Wallis test with Dunn’s multiple comparisons was applied for multiple group comparisons. Fisher’s exact test was used to compare the proportions between groups. Univariate and multivariate analyses of factors that were associated with low immunologic response and late immune response were analyzed. The graphics were created using GraphPad Prism version 9.0 software (Graph-Pad Software, Inc., San Diego, CA, USA). Both STATA version 16.1 (Stata Corp., College Station, TX, USA) and GraphPad Prism version 9.0 software were used for statistical analyses. A two-sided *p*-value < 0.05 was considered to be statistically significant.

## 3. Results

### 3.1. Patient Characteristics

Of the 367 patients who received two doses of the ChAdOx1 nCoV-19 vaccine, 345 (94%) and 290 patients (79%) had an evaluable serological test at 4 weeks and 12 weeks post second dose, respectively. At the data cutoff of 1 March 2022, the median follow-up was 180 days (IQR 169, 190 days). In total, 55 patients were excluded due to consent withdrawal (31 patients), cancer-related death (14 patients), and SARS-CoV-2 infection (10 patients). Thus, 290 participants were included in the analysis for durability outcome (Appendix A). In order to exclude asymptomatic SARS-CoV-2 infections, anti-nucleocapsid testing was performed in all 290 patients, and none were found to be positive for anti-nucleocapsid antibodies.

Of the 290 patients, 186 (64%) were female with a median (IQR) age of 61 years (50–67.75). The majority of enrolled patients had breast cancer (33.45%), lung cancer (26.21%), and colorectal cancer (20.34%). Nearly half of the patients initially received chemotherapy (48.6%), followed by targeted therapy or CDKi (29.3%), and immunotherapy (12%). The disease stages in patients included stage I (5.17%), stage II (16.20%), stage III (34.82%), and stage IV (43.79%) cancers. The majority of patients had ECOG 0–1 (274 out of 290 individuals, 94.48%), and most had no previous significant medical history (70.34%). For the group with comorbidities, cardiovascular diseases, including hypertension, cerebrovascular disease, and coronary artery disease had the highest proportion (45.5%), followed by diabetes (15.9%) and CKD (3.1%) (Table 1).

### 3.2. Anti-RBD Total Ig Response at 12 Weeks Post Second ChAdOx1 nCoV-19 Vaccine

At three months after the second dose, the geometric mean titers (GMTs) of RBD total Ig in cancer patients had rapidly waned by 42%, compared to the GMT at 4 weeks post second dose (GMT 149.7 [95% CI 119–188.3] versus 256.4 BAU/mL [95% CI 199.3–329.7]). In the cancer cohort, 18/290 patients (6.20%) became seronegative, 12 weeks post second dose, compared with 15/290 patients (5.17%), 4 weeks post-second dose. In contrast, no patients from the healthy cohort were seronegative at three months after the second dose.

In comparison with healthy adults, cancer patients showed a lower antibody response by 3.65-fold at 4 weeks post second dose (GMT 256.4 versus 935.4 BAU/mL, *p* < 0.0001), and by 3.22-fold at 12 weeks post second dose (GMT 149.7 versus 482.1 BAU/mL, *p* < 0.0001), respectively (Figure 1).

### 3.3. Decay Rate of Anti-SARS-CoV-2 Ig

Three months after completion of the two-dose series of the ChAdOx-1 COVID-19 vaccine, we assessed the decay rate of anti-SARS-CoV-2 Ig using the slope of the regression line that was adjusted for individual baseline values. In the cancer cohort, we found a decay rate of 0.004155 (log 10 scale)/day, which was not significantly different from the controls (0.004736 (log 10 scale)/day, *p* = 0.244) (Figure 2A,B).

When comparing the decay rate of anti-SARS-CoV-2 Ig across different initial types of treatment, with adjusted baseline values, the decay rate was highest in the immunotherapy group (0.008619 (log 10 scale)/day), followed by the targeted therapy group (0.005772 (log 10 scale)/day), chemotherapy group (0.002552 (log 10 scale)/day), and biologic agent/anti-hormone group (0.002066 (log 10 scale)/day). The differences in the decay rate were found to be statistically significant, when compared to the chemotherapy group, for the targeted therapy/CDK4/6 inhibitor (*p* = 0.0011) and immunotherapy groups (*p* < 0.0001). The immunotherapy group also had a significantly larger decay rate in anti-SARS-CoV-2 total Ig when compared to those of the biologic agents group and the anti-hormonal treatment group. However, no significant differences were found between the chemotherapy and targeted therapy groups vs. the biologic agent/anti-hormone groups, and between the immunotherapy group vs. the targeted therapy/CDK4/6 inhibitor group (Figure 2C,D).

### 3.4. Late Titer Elevation in SARS-CoV-2 Binding Antibody in Solid Cancer Patients

Previous data showed that the peak of the COVID-19 vaccine-induced antibody response occurred at approximately 3–5 weeks after vaccination completion [15]. Interestingly, we observed an increase in the anti-RBD total Ig titers at 12 weeks compared to 4 weeks, post second dose, in a proportion of cancer patients; this event was defined to be a late titer rising event. These patients tested negative for anti-nucleocapsid antibodies. Seventeen percent of cancer patients (50 of 290) had a late titer elevation compared with 0.01% (1 of 84) in the healthy subjects.

In order to predict which factors were correlated with this event, univariate and multivariate analyses were performed. In the univariate analysis, breast cancer (OR 3.99, [95% CI: 1.54–10.35], *p* = 0.0043), chemotherapy treatment (OR 4.51, [95% CI: 1.81–11.24], *p* = 0.0012), biologic agent/anti-hormonal treatment (OR 3.43, [95% CI: 1.81–11.24], *p* = 0.0012), and steroid use (OR 2.18, [95% CI: 1.15–4.13], *p* = 0.0017) were associated with late immunologic responses. Multivariate regression analysis, based on the significant factors that were identified from the univariate analysis, showed that the chemotherapy treatment group was the only factor that retained significance with late immunologic response (OR 4.31, [95% CI: 1.1–16.96], *p* = 0.036) (Appendix A). The chemotherapy group had a significantly higher proportion of patients with late titer elevation than that in the healthy group. However, there was no difference in the proportion of patients with late titer elevation between other treatment types and healthy controls (Appendix A).

### 3.5. Clinical Factors Associated with Adequate and Non-Adequate Immunologic Response

Currently, there are no agreed upon standardized cut-off values for anti-RBD total Ig and neutralizing antibody response to assess vaccine effectiveness against SARS-CoV-2. Many studies set the cut-off value as the seronegative rate, while others use the value of the convalescent plasma [7,9,16]. We selected a cut-off value of 210 U/mL (Elecsys Anti-SARS-CoV-2 S), which was acceptable for use in the manufacture of COVID-19 convalescent plasma with high titers of anti-SARS-CoV-2 antibodies issued by the Food and Drug Administration (FDA) as an adequate response level [14].

In the univariate logistic regression analysis, the statistically significant predictors of inadequate response included male gender (OR 2.10; 95% CI: 1.24, 3.57, *p* = 0.0061), comorbidity (OR 2.12; 95% CI: 1.24, 3.62, *p* = 0.0057), and immunotherapy as an initial treatment (OR 4.24; 95% CI: 1.81, 9.91, *p* = 0.0009). The multivariate regression analysis found significant associations between immunotherapy treatment (OR 2.23, [95% CI: 1.31–7.94], *p* = 0.011) and comorbidity (OR 1.90, [95% CI: 1.08–3.35], *p* = 0.026), with an inadequate immunologic response (Appendix A).

### 3.6. Effect of Anticancer Treatment on Antibody Response

Of the 290 patients, 137 patients continued the same anticancer therapy from the first vaccine dose to 12 weeks post second dose. In order to assess the unmixed effect of anticancer treatment, only those who continued the same therapy were analyzed.

The follow-up analysis at three months showed significantly lower immunogenicity levels in patients who underwent different types of treatment compared to the healthy controls.

At 12 weeks post second dose, the GMTs of anti-RBD total Ig were 55.97 BAU/mL (95% CI 23.16, 135.2), 106.4 BAU/mL (95% CI 54.58, 207.4), and 195.2 BAU/mL (95% CI 142.6, 267.2), in the immunotherapy, chemotherapy, and targeted therapy groups, respectively (Figure 3).

Among the three types of treatment groups, we found that the chemotherapy and immunotherapy groups had a significantly higher proportion of patients in the low titer group than those in the targeted therapy and healthy groups (Figure 3). Patients in the targeted group also had a higher proportion of patients in the non–adequate response group than that in the healthy group. Conversely, the proportion of patients who had non-adequate responses was not significantly different between those in the immunotherapy and chemotherapy groups (Appendix A).

### 3.7. Effect of Treatment Cessation on Antibody Response

In order to explore the effect of treatment cessation on antibody response, patients were categorized into two groups based on discontinuation of treatment for at least 4 weeks before antibody testing at 3 months post second dose. Of the 290 patients, there were 227 patients undergoing any type of cancer treatment, and 63 patients who discontinued treatment. At both 1 month and 3 months after the second dose, the anti-RBD total Ig levels in patients who ceased treatment were numerically higher than those undergoing cancer treatments; however, this was statistically insignificant (Figure 4A). The decay rate of patients who ceased treatment was not statistically different from those who did not cease treatment (0.003885 vs. 0.004224 (log 10 scale)/day, *p* = 0.83) (Figure 4B,C).

### 3.8. Surrogate Neutralization against Omicron Variant of Concern

Based on 40 of the 290 serum samples with a high level of anti-RBD total Ig among the four types of treatment, only two samples tested positive. Of all 40 samples, 37 were in the adequate response group (defined as anti-RBD total Ig > 210 U/mL), and only 2 of these 37 patients had detectable neutralizing antibodies against the Omicron BA.2 variant (Table 2).

## 4. Discussion

Consistent with previous studies of the immune response after two doses of mRNA COVID-19 vaccines in cancer patients [9,10,17], we found that the humoral immunity of vector-based vaccine protection in both cancer and healthy cohorts dropped over time. Although a high proportion of patients (93%) remained seropositive 3 months post second dose, the levels of anti-RBD total Ig decreased by 42%, significantly more than those in the healthy controls. Of the 367 cancer patients, 12 (3.3%) who received the complete two doses had COVID-19 infections compared to 1% (1/90) COVID-19 infection found in healthy controls. Despite the initially lower levels anti-RBD total Ig in the cancer cohorts, their decay rates were not significantly different from that in the healthy cohort.

Among different cancer treatment types, the immunotherapy and chemotherapy groups had a significantly higher proportion of patients with a lower immune response than the targeted therapy and healthy groups. It is worth noting that patients on immunotherapy had the lowest antibody levels, with GMT of 55 BAU/mL (95% CI 23.16–135.2), which was even lower than that of chemotherapy group. This evidence supports the notion that both immunotherapy and chemotherapy can blunt immunity in cancer patients receiving the SAR-CoV-2 vaccine.

In multivariate analysis, immunotherapy, and the presence of any comorbidities were associated with inadequate antibody responses, defined to be lower than 210 U/mL. Although chemotherapy was a well-known factor associated with a lower level of mRNA vaccine-induced immune response [18], patients receiving chemotherapy were not associated with diminished immune responses in the multivariate analysis. This contrasting result between our results and those of previous studies could be partly due to differences in the cancer stage and cessation of the treatment. In this study, 70 of 98 patients in the chemotherapy group had early-stage cancer, and used chemotherapy as an adjuvant. Eventually, their treatment was discontinued within the three-month time point following completion of the two-dose vaccination. In contrast, the majority of patients in the immunotherapy group were diagnosed with advanced stage cancer, and received immunotherapy as a late-line treatment.

It is critical to find a balance between increasing the antiviral immune response, while reducing the inflammatory response and other complications from cancer. One of the major issues in public health is whether systemic cancer treatment should be discontinued in order to improve the vaccine-induced immune response, thereby protecting cancer patients from SARS-CoV-2-related death and severe clinical symptoms. Our results support the case that the group that ceased cancer treatment exhibited an improved antibody response compared to groups that did not cease treatment. The differences in the responses that followed cessation of chemotherapy, immunotherapy, and targeted therapy, were not statistically different. It is possible that some patients who discontinued the drug may have experienced disease progression or poor-performance status, both of which potentially affect immunogenicity and durability of the humoral response. Another confounding factor was the differing intervals of time following drug discontinuation during the follow-up period. We included both patients who stopped treatment after the first vaccination and after the second vaccination in the treatment cessation group. The response patterns may become more evident in a longitudinal study that extends the follow-up period; however, we may not be able to perform this analysis due to recent implementation of the 3-dose booster policy.

One of the key findings in our study was that 50 of 290 (17%) patients in the cancer cohort had late titer elevations in immune response, which was extremely rare in the healthy control population. A previous SARS-CoV-2 infection may explain the late antibody formation; however, we controlled for previous infection with results from the nucleocapsid test. Findings from the multivariate analyses revealed that chemotherapy was an important factor associated with the late titer elevation in immune response. A probable explanation for this finding is that some patients in the chemotherapy group may have stopped or changed their treatment regimen to a less stressful therapy, such as in the adjuvant setting of breast cancer.

In the context of Omicron, which is currently the widespread variant of concern, it has been reported that this variant has the potential to evade vaccine-induced humoral immunity in healthy populations [19]. In the general population, the recommendation from the WHO for additional doses is 4–6 months after primary vaccination series completion, especially for the Omicron variant [20]. In immunocompromised hosts, the recommendation is currently to shorten the duration of time for the subsequent dose. For cancer patients, an additional dose is recommended up to 4 months after the primary vaccine series, for both the mRNA and viral vector vaccine series, because the immunogenicity after two doses may not provide adequate protection [21]. Our results were compatible with this recommendation. An increase in an antibody titer following a two-dose vaccination appears to be ineffective against the Omicron variant. Many studies have attempted to determine an absolute threshold for the cut-off value for the anti-RBD level that corresponds with neutralization against wild type (Wuhan) and Delta strains [22], but there are limited data on the neutralization of the Omicron spike by antibody, which is expected to correspond to a far higher titer level. Even with the high-titer group in our cancer cohort, the neutralizing activity against Omicron was deemed poor. For example, in the adequate response group, only 1 of 4 patients tested positive for sVNT against Omicron BA.2. Previous data on various types of vaccine regimens suggested that an extended duration between additional doses could increase the antibody response [23,24,25]. However, the longer the wait, the greater the risk of infection becomes during that period of time. This fact underscores the importance of administering booster doses at 3 months or earlier, especially in oncologic patients during the Omicron pandemic.

There are some limitations in our report. Firstly, during the time period when we were conducting this research, Thailand was in the midst of the Omicron BA.2 pandemic. In the near future, the Omicron BA.5 variant is likely to become the main spreading strain. Secondly, we did not perform a cellular immunity test to evaluate vaccine effectiveness against Omicron BA.2. Thirdly, the sVNT against wild type and a previous variant of concern, such as alpha and beta variants, was not tested; moreover, a limited number of serum samples were evaluated for neutralization against Omicron BA.2. However, we can only speculate on the results of the non-selected samples with lower titer, while the sVNT results are expected to be negative.

Our results have helped to fill in knowledge gaps regarding the durability of the immune response after the viral vector-based vaccine, which was widely administered across Thailand prior to the mRNA-based vaccine. Obtaining data on the immune response in immunocompromised groups such as cancer patients with varying treatment regimens, has become a priority in informing the design of vaccination programs that can improve protection against emerging COVID-19 variants.

## 5. Conclusions

Waning immunogenicity was observed three months after completing ChAdOx1 nCoV-19 vaccination in a solid cancer patient cohort undergoing anticancer treatment, although the seroconversion rate was maintained. Among the treatment subgroups evaluated, the immunotherapy treatment group produced a lower immunogenic response and had a more significant decline in anti-SARS-CoV-2 Ig antibodies. Chemotherapy treatment was associated with a unique, late immunologic response. Most importantly, in oncologic patients, immunogenicity after 3–4 months of completing the ChAdOx1-nCoV-19 vaccine series was not sufficient for protection against SAR-CoV-2 Omicron BA. 2. An additional dose must be administered in this vulnerable population.

## Figures and Tables

**Figure 1 vaccines-10-01662-f001:**
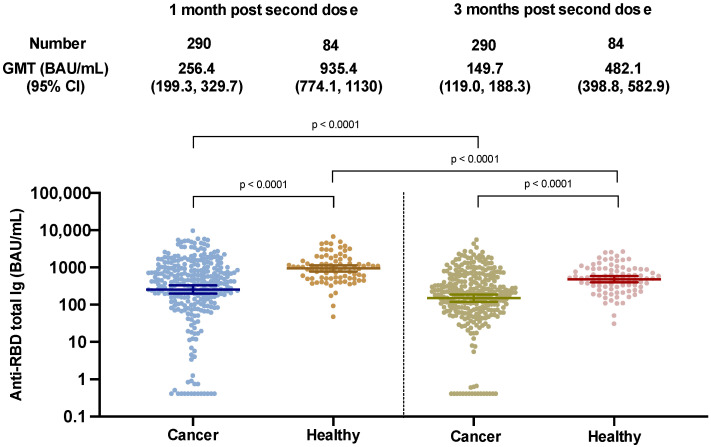
SARS-CoV-2 binding antibody levels at 1 and 3 months after two doses of the ChAdOx-nCoV-19 vaccine in cancer and healthy cohorts.

**Figure 2 vaccines-10-01662-f002:**
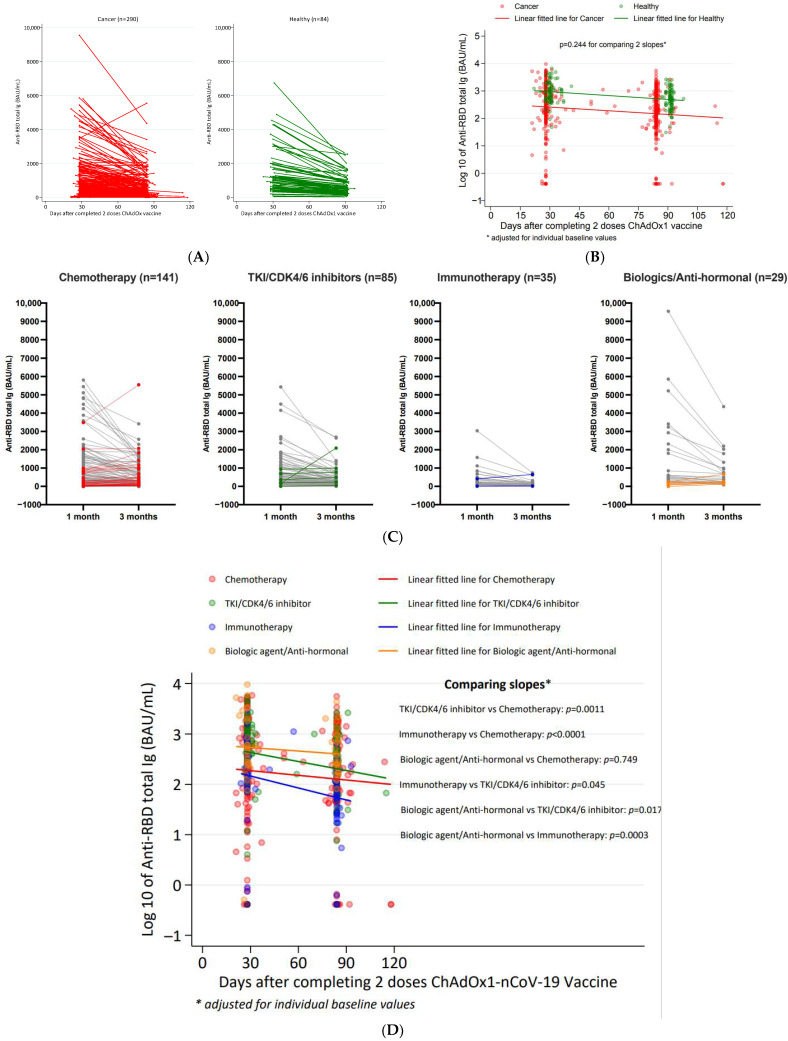
Decay rates of SARS-CoV-2 binding antibody levels at 1 and 3 months after two doses of the ChAdOx-nCoV-19 vaccine in cancer (n = 290) and healthy cohorts (n = 84). (**A**) Comparison of anti-RBD total Ig at 3 months after completing 2 doses of ChAdOx1-nCoV-19 between the cancer and healthy groups—adjusted for individual baseline values (at 1 month). (**B**) Slopes of regression lines adjusted for individual baseline values (at 1 month): slope (cancer) = −0.004155 (log 10 scale)/day, slope (healthy) = −0.004736 (log 10 scale)/day. (**C**) Comparison of anti-RBD total Ig at 3 months after completing 2 doses of ChAdOx1-nCoV-19 between treatment types among cancer patients—adjusted for individual baseline values (at 1 month). Grey line represents patients with decreasing antibody levels and colored lines represent patients with stable or rising levels. (**D**) Slopes of regression lines adjusted for individual baseline values (at 1 month): slope (chemotherapy) = −0.002552 (log 10 scale)/day, slope (TKI/CDK4/6 inhibitor) = −0.005772 (log 10 scale)/day, slope (immunotherapy) = −0.008619 (log 10 scale)/day, slope (biologic agent/anti-hormonal) = −0.002066 (log 10 scale)/day. * In order to consider multiple comparisons adjustment using Bonferroni, the calculated *p*-values should be compared with <0.05/6 = <0.0083 in considering significance.

**Figure 3 vaccines-10-01662-f003:**
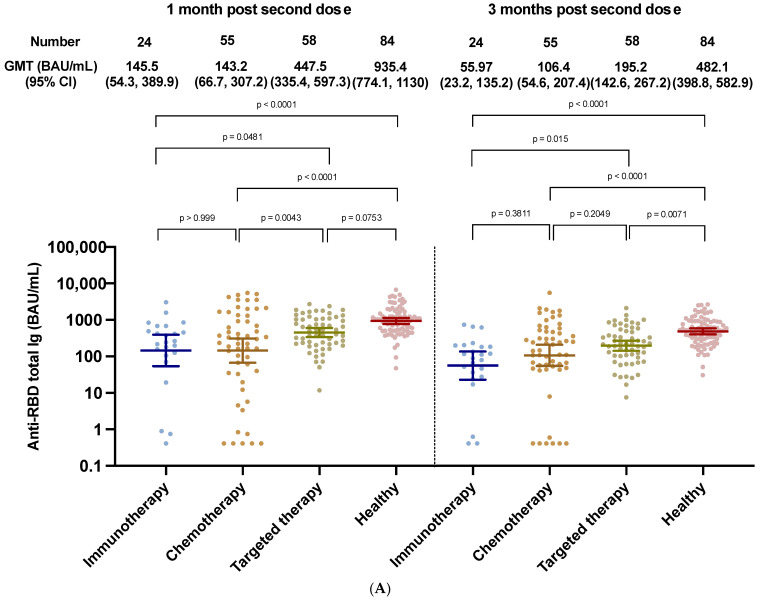
SARS-CoV-2 binding antibody levels at 1 and 3 months after two doses of the ChAdOx-nCoV-19 vaccine in cancer patients stratified by treatment given (n = 137) compared to healthy controls (n = 84) (**A**). GMT of Anti-RBD total Ig of SAR-CoV-2 at one month and three months after completing ChAdOx1-nCoV-19 vaccines among different types of anticancer treatments (**B**): immunotherapy, chemotherapy, targeted therapy.

**Figure 4 vaccines-10-01662-f004:**
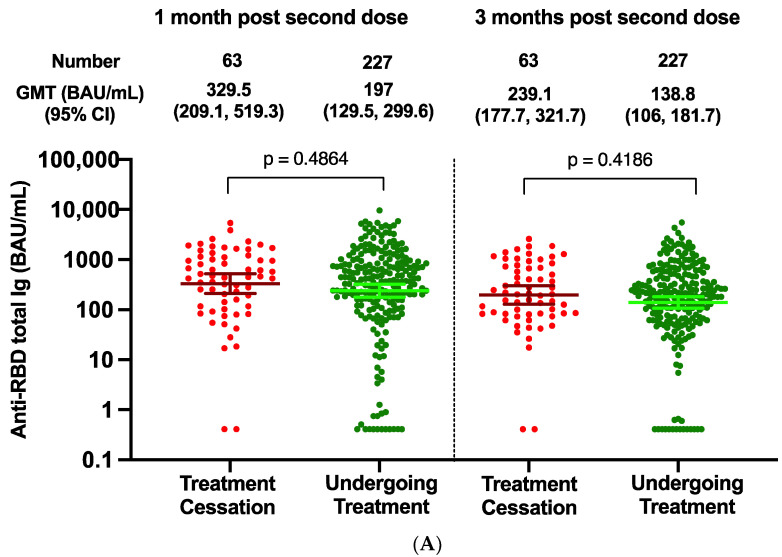
Effect of treatment cessation on antibody response. The SARS-CoV-2 binding antibody levels at 1 and 3 months after two doses of the ChAdOx-nCoV-19 vaccine in cancer treatment cessation patients (n = 63, red color), and those undergoing anticancer treatment (n = 227, green color) (**A**). Dynamics (**B**) and slopes (**C**) of anti-RBD total Ig at 1 month and 3 months after the second dose of ChAdOx1-nCoV-19 among cancer patients who discontinued treatment compared to those receiving anticancer treatment—adjusted for individual baseline values (at 1 month). Slopes of regression lines adjusted for individual baseline values (at 1 month): slope (treatment interruption) = −0.003885 (log 10 scale)/day, slope (no interruption) = −0.004224 (log 10 scale)/day.

**Table 1 vaccines-10-01662-t001:** Demographic data of 290 oncologic patients.

**Age, Years, Median (IQR)**	61 (50, 67.75)	**Cancer type**	
**BMI ± SD**	22.98 ± 4.20	Breast	97 (33.45%)
**Sex**		Lung	76 (26.21%)
Female	186 (64.14%)	Colorectal	59 (20.34%)
Male	104 (35.86%)	GIST	11 (3.79%)
**ECOG**		Head and neck	8 (2.76%)
ECOG 0–1	274 (94.48%)	Cholangiocarcinoma	2 (0.69%)
ECOG 2	16 (5.51%)	Others	37 (12.76%)
**Initial TNM staging**		Cancer treatment within 4 weeks before first vaccination	
I	15 (5.17%)	Chemotherapy	141 (48.6%)
II	47 (16.20%)	Oxaliplatin-containing regimen	34
III	101 (34.82%)	Anthracycline	24
IV	127 (43.79%)	Plalinum doublet	23
**Comorbidities**		Paclitaxel	18
Cardiovascular disease *	95 (45.46%)	5-FU or Gemcitabine	15
Diabetes	46 (15.86%)	Irinotecan-containing regimen	15
COPD	4 (1.38%)	Docetaxel	7
Cirrhosis	8 (2.76%)	Other	5
CKD	9 (3.10%)	Targeted therapy/CDKi	85 (29.3%)
Others	53 (18.28%)	Immunotherapy	35 (12%)
No comorbidities	204 (70.34%)	Single agent anti-PD1/PDL1	33
**Time to blood collection from second dose**		Anti-PD1 plus nati-CTLA4	3
**4 weeks post second dose (Days ± SD)**	28.86 ± 6.49	Biologic agent	16 (5.5%)
**12 weeks post second dose (Days ± SD)**	86.46 ± 22.78	Anti-hormonal treatment	13 (4.5%)

* Cardiovascular disease also includes hypertension, cerebrovascular disease, and coronary artery disease.

**Table 2 vaccines-10-01662-t002:** Surrogate neutralization for SARS-CoV-2 Omicron (B.1.1.529) BA. 2 in solid cancer patients.

		Neutralization against Omicron (30% Cut-Off)	
		Positive	Negative
**Anti-RBD Ig > 210 U/mL**	Adequate response	2	35
**Anti-RBD Ig < 210 U/mL**	Inadequate response	0	3
	Total	2	38

## Data Availability

Data are available upon reasonable request to the corresponding author.

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
