# Peer review of "Durability of Immune Response to ChAdOx1-nCoV-19 Vaccine in Solid Cancer Patients Undergoing Anticancer Treatment"

_vaccines, 2022, doi:10.3390/vaccines10101662_

Round 1

Reviewer 1 Report

Overall this paper is providing great information to physician especially in medical oncology field. 

the paper is well structured, and The title of this paper is very interesting and clear. and the author has been able to elaborate and discussed the purpose of this article through out his manuscript.

The objectives are seems to be clear and is going to significantly improve the clinical practice especially in this COVID 19 eras.

The discussion of the paper appear to be very cohesive, the evaluating methods seems to be well rounded.

I think I personally congratulate the authors for their effort to write this paper. 

being said that, i would like to suggest to the authors if they are able to include "immunocompromise patients especially organ transplant patients whom they are on immunosuppressive medication in this study.

Author Response

We thanks the reviewer for the comment.

Reviewer 2 Report

General comments. This is an observational, longitudinal -quite accurate- study of the antibody response at 3 months after a second dose of ChAdOx1-nCov19 vaccine in 290 patients with different solid cancer under different treatment schedules, compared to antibody response in 84 healthy individuals. The authors already reported on the one-month response in the same cohort. Now, after some preliminary observations, they extend the evaluation to the extent and kinetics of the antibody response after a longer observation period . The entity and durability of the immune reponse after different COVID-19 vaccines in cancer patients has been investigated in several studies. However, the majority of studies deal with nucleic acid vaccines while in some countries only viral vectors vaccines were available. Therefore, there is some interest in these countries for data presented in the article, even if their novelty is limited. Moreover, there is already a substantial consensus on the need of a booster dose  with any vaccine. Accordingly, data after this already widely implemented vaccination schedule would have been of more interest. Methods and analysis of data are accurate. Perhaps the text and figures are redundant in relation to the findings.

Minor comments. Introduction,lines 43-45. There should be a comma between the two sentences. Lines 45-49. The sentence should be revised.

Author Response

Thank you very much for your comments

Minor comments. Introduction,lines 43-45. There should be a comma between the two sentences.

A common was added Line 46-47

Lines 45-49. The sentence should be revised.

The sentence has been revised Line 48-49

Reviewer 3 Report

In this prospective longitudinal observational study, the authors compared the durability of the humoral immune response (binding total Ig) to the widely used adenovirus-vectored ChAdOx1-nCoV-19 vaccine in 290 solid cancer patients with different active anti-cancer treatments and 90 healthy individuals at 4 and 12 weeks after the second dose of vaccine at two cancer centers in Thailand.  Particularly with chemotherapy and immunotherapy, the waning of immunogenicity observed within 3 months after the second dose of vaccine indicates that the vaccine likely did not provide sufficient neutralization protection against SARS-CoV-2 Omicron BA.2 variant.

This study appears to have been well conducted and analyzed, and provides a rationale for a booster dose of vaccine for cancer patients.

Line 27: define all acronyms at their first mention in the abstract and the main text. Is GMT geometric mean titer?  At line 104, S/C ratio?

Line 39-40, 46-49: what was the magnitude of the changes described? Please include quantitative information from the literature.  Mention that the mRNA vaccine was BNT162b2.

Line 54-61: this paragraph could be inserted after line 49-50 to elaborate on what is known about the vaccine response in cancer patients.  Cite the references of your previous work (ref. 11, 12).

Table 1: there’s no need to repeat “/290” throughout the list of results. The total number of cancer patients is in the title of the table.  However, please provide more detail about the cancer types (most common pathologic diagnoses) and initial cancer treatments (names of the most common therapeutic agents, particularly for chemotherapy and immunotherapy).  What was the timing relationship between the second dose of vaccine and the treatment regimens?

Line 241-243, Fig. 3: clearly label each panel and describe each panel separately in the legend.

Line 259: not “a patient”, but patients.

Line 270-272, Fig. 4: be consistent with the size of the panels and figures throughout the paper.  Clearly label each panel and describe each panel separately in the legend.  How long was the treatment cessation for each type of treat

Line 280: isn’t that Fig. 5?

Line 287: do you mean “Consistent with the results of the immune response…”?

Line 305-313: which chemotherapy?  Which immunotherapy?

Line 335-337: have you correlated the type and timing of cancer treatment regimens with the presence of late antibody formation in individual patients?

Reviewer 4 Report

1. The authors state in the paragraph of lines 300-302 the following: "This supports that both immunotherapy and chemotherapy are important factors for blunting immunity in cancer patients receiving the SAR-CoV-2 vaccine"; however, they do not establish the references that support this conception. Therefore, the authors should cite those articles that support this idea.

2. Reference 18 does not state chemotherapy anywhere. This study (18) included 43,548 patients, none of whom were identified as having cancer, much less receiving chemotherapy. Therefore, it is wrong to cite reference 18. The authors should correct this citation and place more appropriate references in relation to the effect of immunotherapy and chemotherapy on the lower Ig response induced by vaccination.

3. The authors should discuss their results with respect to those obtained by Guven DC et al (DOI: 10.1002/ijc.34280), Teeyapun et al (DOI: 10.1016/j.eclinm.2022.101608), and Mekkawi et al (DOI: 10.1177/ 10732748221106266).
